# Dietary Fermentation Product of *Aspergillus Oryzae* Prevents Increases in Gastrointestinal Permeability (‘Leaky Gut’) in Horses Undergoing Combined Transport and Exercise

**DOI:** 10.3390/ani13050951

**Published:** 2023-03-06

**Authors:** Melissa McGilloway, Shannon Manley, Alyssa Aho, Keisha N. Heeringa, Lynsey Whitacre, Yanping Lou, E. James Squires, Wendy Pearson

**Affiliations:** 1Department of Animal Biosciences, University of Guelph, Guelph, ON N1G 2W1, Canada; 2BioZyme Inc., St. Joseph, MO 64504, USA

**Keywords:** leaky gut syndrome, horses, hyperpermeability, prebiotics

## Abstract

**Simple Summary:**

Equine leaky gut syndrome is characterized by gastrointestinal hyperpermeability and may be associated with adverse health effects in horses. The purpose was to evaluate the effects of a prebiotic *Aspergillus oryzae* product (SUPP) on the stress-induced leakiness of the gut. For 28 days, 8 horses received a diet containing the prebiotic or an unsupplemented diet (CO). On Days 0 and 28, horses were dosed with a compound (iohexol) that should only leak out of the gastrointestinal tract if the gut walls become leaky. Immediately following iohexol administration, four horses from each feeding group underwent 60 min of transport immediately followed by a moderate-intensity exercise bout of 30 min (EX), and the remaining horses were maintained as sedentary controls (SED). Blood was sampled before iohexol, immediately after trailering, and at 0, 1, 2, 4, and 8 h post-exercise. Blood was analyzed for iohexol, as well as lipopolysaccharide (a compound found in the gastrointestinal tract that can leak out) and serum amyloid A (a marker of inflammatory response). EX resulted in a significant increase in plasma iohexol in both CO and SUPP groups on Day 0; this increase was not seen in SED horses. On Day 28, EX increased plasma iohexol only in the CO feeding group; this increase was completely prevented by the provision of SUPP. It is concluded that combined transport and exercise induce leaky gut. Dietary SUPP prevents this and therefore may be a useful prophylactic for pathologies associated with gastrointestinal hyperpermeability in horses.

**Abstract:**

Equine leaky gut syndrome is characterized by gastrointestinal hyperpermeability and may be associated with adverse health effects in horses. The purpose was to evaluate the effects of a prebiotic *Aspergillus oryzae* product (SUPP) on stress-induced gastrointestinal hyperpermeability. Eight horses received a diet containing SUPP (0.02 g/kg BW) or an unsupplemented diet (CO) (n = 4 per group) for 28 days. On Days 0 and 28, horses were intubated with an indigestible marker of gastrointestinal permeability (iohexol). Half the horses from each feeding group underwent 60 min of transport by trailer immediately followed by a moderate-intensity exercise bout of 30 min (EX), and the remaining horses stayed in stalls as controls (SED). Blood was sampled before iohexol, immediately after trailering, and at 0, 1, 2, 4, and 8 h post-exercise. At the end of the feeding period, horses were washed out for 28 days before being assigned to the opposite feeding group, and the study was replicated. Blood was analyzed for iohexol (HPLC), lipopolysaccharide (ELISA), and serum amyloid A (latex agglutination assay). Data were analyzed using three-way and two-way ANOVA. On Day 0, the combined challenge of trailer transport and exercise significantly increased plasma iohexol in both feeding groups; this increase was not seen in SED horses. On Day 28, EX increased plasma iohexol only in the CO feeding group; this increase was completely prevented by the provision of SUPP. It is concluded that combined transport and exercise induce gastrointestinal hyperpermeability. Dietary SUPP prevents this and therefore may be a useful prophylactic for pathologies associated with gastrointestinal hyperpermeability in horses.

## 1. Introduction

Leaky gut syndrome (LGS) is characterized by gastrointestinal hyperpermeability and increased accessibility of the systemic environment to compounds that are normally sequestered within the gastrointestinal lumen [1]. The contribution of LGS to equine disease is poorly understood, and its mitigation by dietary interventions has not been described in the literature. An MSc thesis from Michigan State University [2] describes a study in which oral phenylbutazone contributed to the development of gastrointestinal hyperpermeability in 18 Arabian horses, suggesting that gastric ulceration, phenylbutazone administration, or both, contribute to the development of LGS in horses. Evidence also implicates diets high in starch as complicit in gastrointestinal hyperpermeability [3]. Exercise is another likely candidate as an LGS risk factor but has not been clearly described in horses. Research in humans, however, provides evidence for a positive correlation between exercise intensity/duration and hyperpermeability of the gastrointestinal tract [4,5,6]. A recent study in eight horses reports that the combination of exercise and trailer transport induces an increase in gastrointestinal permeability, as well as increased serum amyloid A and lipopolysaccharide [7]. Whilst the pathophysiological consequences of LGS are as vaguely characterized as its triggers, there is evidence that, depending on the degree of inflammatory response to luminal toxins, LGS may impair skeletal muscle metabolism [8], and contribute to metabolic dysfunction [9,10], allergies [11,12], and inflammatory diseases such as arthritis [13]. Dietary interventions with evidence for an ability to protect against the development or clinical consequences of LGS will make an important contribution to preserving robust equine health.

Perhaps due (at least in part) to the incomplete picture defining the cause-and-effect of LGS, interventions tend to rely heavily on the management of downstream clinical consequences. To the authors’ knowledge, there are currently no feed supplements or pharmaceutical drugs that have been evaluated against the gastrointestinal hyperpermeability that is the cornerstone of LGS. A commonly reported feature of LGS in non-equine species is gastrointestinal dysbiosis, and there is evidence that this dysbiosis contributes to the development of hyperpermeability [14,15,16,17]. Dysbiosis is likely in horses receiving a high-starch diet [3,16], and in horses experiencing physiological stress [16]. Thus, interventions with potential to stabilize gastrointestinal microbiota may protect against the development of hyperpermeability under conditions of stress. *Aspergillus oryzae* is a filamentous fungus, which has demonstrated the ability to amplify the abundance of probiotic microbes (particularly *Bifidobacterium pseudolongum*) whilst protecting DSS-challenged mice against colitis [18]. The fermentation product of *A. oryzae* also promotes fiber-degrading bacteria in the rumen and hindgut when fed to lactating dairy cows [19]. In addition to evidence for a prebiotic-like effect, *A. oryzae* also exerts a marked anti-inflammatory effect in LPS-stimulated polymorphonuclear cells and improves the structure of gastrointestinal lumen (i.e., villus height–crypt ratio) in broiler chickens [20]. Furthermore, the administration of a postbiotic from *A. oryzae* to calves prevented the increase in intestinal permeability associated with exposure to high ambient temperature [21]. These data support the hypothesis that *A. oryzae* protects against stress-induced hyperpermeability by amplifying the abundance of a healthy gastrointestinal microbiome. Accordingly, the purpose of the current study was to evaluate the effects of a fungal prebiotic produced through a proprietary fermentation process with *A. oryzae* (SUPP; BioZyme Inc.; St. Joseph, MO, USA) on equine gastrointestinal hyperpermeability induced by a combination of trailer transport and moderate-intensity exercise horses. The objectives were to characterize the effect of a dietary *A. oryzae* prebiotic on the appearance and disappearance of an oral permeability marker (iohexol) in the blood of horses challenged with combined transport and exercise stress, and to correlate observed effects with those on downstream evidence of inflammation (serum amyloid A (SAA)) and translocation of enteric endotoxin (lipopolysaccharide (LPS)).

## 2. Materials and Methods

Care and use of animals was reviewed and approved by the University of Guelph Animal Care Committee in compliance with the guidelines published by the Canadian Council on Animal Care (Approval Number 3800).

### 2.1. Horses

Eight (8) healthy mares (Age: 14.2 ± 3.7 years; body weight: 570 ± 47.4 kg) from the Arkell Equine Research Station, University of Guelph, were included in the randomized, partial cross-over trial. The horses were group-housed in an open turnout area, with unrestricted access to a large covered shelter bedded with straw, 1st cut Timothy hay, water, and trace mineral salt. Two hundred and fifty (250) g of a 12% maintenance pellet ration^a^ was provided once per day (morning) (Table 1). Horses were all accustomed to a lifestyle that did not include forced exercise. 

At the beginning of the study, all 8 horses were randomized into one of two feeding groups (n = 4 per group): Group A: unsupplemented control diet (CO); Group B: diet containing *A. oryzae* prebiotic^b^ (SUPP; 0.02 g/kg BW). SUPP was a textured, unpelleted product and was top-dressed onto the horse’s individual pelleted feed once per day. Horses consumed their pelleted ration with or without SUPP once per day in individual stalls. Once their feed was completely consumed, they were returned to the outdoor turnout area. Within each feeding group, horses were further divided into stress-challenged (EX—see below for details) or non-challenged sedentary controls (SED) (n = 2 per group per replicate). Horses received their assigned diet for 28 days. On Days 0 and 28, one SED and one EX horse were evaluated in the morning, and a second SED and second EX horse were evaluated in the afternoon. At the end of the 28-day feeding period, horses were washed out for 28 days, and then assigned to the opposite feeding group for an additional 28 days. The trial was then repeated, for a final ‘n’ of 8 per feeding group (i.e., 4 × EX and 4 × SED per feeding group). Horses were tested at the same time of day (morning or afternoon) in both study periods.

On study days, horses remained in their turnout area with unrestricted water access, but from which all feed had been removed. Following 12 h of fasting, horses were stalled and administered via nasogastric tube an indigestible marker of gastrointestinal permeability (iohexol^c^; 5.6% solution, 1.0 mL/per kg BW; 56 mg/kg BW) by a licensed veterinary professional [7]. The procedure was conducted in the absence of any sedation, so as not to interfere with normal gastrointestinal motility [22].

### 2.2. Stress Challenge

Horses were challenged with combined trailer transport and exercise, which we have previously demonstrated to produce a measurable and significant increase in gastrointestinal hyperpermeability [7]. Briefly, following the administration of iohexol, one EX horse was walked onto a 2-horse trailer for a 60 min drive to the Equine Sports Medicine and Reproduction Centre, University of Guelph. Once at the facility, a heart rate (HR) monitor^d^ was attached to the horse using a flexible belly-band, and the horse was free-lunged around an indoor arena (5 min’ walk, 10 min trot (left), 10 min trot (right), and 5 min’ walk) on a sand footing for 30 min. Horses were encouraged to achieve an exercise intensity that resulted in a HR of approximately 150 bpm during the trot, in order to encourage the horse to work at or beyond the anaerobic threshold [23]. At the cessation of exercise, EX horses returned to the group housing yard directly and were turned out with unrestricted access to hay and water. This challenge has previously been demonstrated to produce gastrointestinal hyperpermeability in horses [6].

Following the application of topical lidocaine at the jugular groove, blood was sampled from the jugular vein immediately before iohexol administration (P1), immediately after trailering (P2), immediately after exercise (P3), and then 1 (P4), 2 (P5), 4 (P6), and 8 h (P7) post-exercise. Blood samples were cooled on ice, centrifuged within 2 h of collection, and the recovered plasma was frozen (−20 °C) until analysis. 

Manure samples were collected within 2 min of voiding before the horse walked into the trailer, at the end of 60 min of transport, and the first manure after exercise. 

### 2.3. Non-Challenged Controls

SED horses received iohexol at the same time as the EX horses, and blood was sampled at the same time as the EX horses. After receiving iohexol they were returned to the group housing area with free access to water. Hay was provided upon return of the EX horse from transport and exercise. 

### 2.4. Sample Analysis

All chemicals and reagents were purchased from Sigma Aldrich^f^, unless otherwise stated. Plasma samples were analyzed for systemic inflammation (serum amyloid A and lipopolysaccharide (LPS)) biomarkers, and an exogenous marker of gastrointestinal permeability (iohexol). 

Plasma iohexol was determined via HPLC (Agilent 1200 series HPLC gradient system), which was used to quantify plasma iohexol (μ g/mL) with UV detection at 254 nm, as previously described [7] (intra- and inter-assay CV: 3.106 and 4.217%, respectively). 

SAA was determined by Eiken Serum Amyloid A latex agglutination assay at a commercial laboratory (Animal Health Laboratory, University of Guelph). 

Plasma samples, acclimated at room temperature, were analyzed in duplicate for LPS (pg/mL) using an equine-specific quantitative sandwich ELISA kit according to manufacturer^h^ instructions (inter- and intra-assay coefficient of variability: 1.5 and 1.6%, respectively). A standard curve was used to generate a linear regression equation, which was used to calculate LPS concentrations in each sample.

### 2.5. Data Analysis

Data analysis was conducted using SigmaPlot^i^ (Version 14.2). Data are presented as mean ± SD unless otherwise indicated. Normality of data was determined using the Shapiro–Wilk test. Three-way ANOVA was used to detect interactions between feeding groups, stress challenge, and time after iohexol administration. Two-way ANOVA was used to identify significant differences between feeding groups in SED and EX horses on Day 0 and Day 28 with respect to stress challenge and time after iohexol administration. The Holm–Sidak post-hoc test was used to identify significantly different means when a significant F-ratio was calculated. Significance was accepted at *p* < 0.05. 

## 3. Results

### 3.1. Gastrointestinal Barrier Function

#### 3.1.1. Control Diet (Figure 1)

Day 0: In SED horses receiving the CO diet, there was no significant change in plasma iohexol at any time between P1 (0.56 ± 0.02 ug/mL) and P7 (0.69 ± 0.04 ug/mL) (*p* = 0.26). EX horses demonstrated a significant increase in plasma iohexol between P1 (0.52 ± 0.03 ug/mL) and P3 (1.14 ± 0.08 ug/mL) (*p* = 0.02). Plasma iohexol was significantly higher in EX horses than in SED horses at P2 (SED: 0.71 ± 0.06 ug/mL; EX: 1.02 ± 0.18 ug/mL) (*p* = 0.04) and P3 (SED: 0.75 ± 0.09 ug/mL; EX: 1.14 ± 0.08 ug/mL) (*p* = 0.01) (Figure 1). 

Day 28: In SED horses receiving the CO diet, there was no significant change in plasma iohexol at any time between P1 (0.48 ± 0.04 ug/mL) and P7 (0.60 ± 0.06 ug/mL) (*p* = 0.44). EX horses demonstrated a significant increase in plasma iohexol between P1 (0.58 ± 0.09 ug/mL) and P3 (1.07 ± 0.06 ug/mL) (*p* = 0.006). Plasma iohexol was significantly higher in EX horses than in SED horses at P2 (SED: 0.54 ± 0.06 ug/mL; EX: 1.01 ± 0.12 ug/mL) (*p* < 0.001), P3 (SED: 0.56 ± 0.07 ug/mL; EX: 1.07 ± 0.12 ug/mL) (*p* < 0.001) and P4 (SED: 0.59 ± 0.04 ug/mL; EX: 1.00 ± 0.10 ug/mL) (*p* < 0.001) (Figure 1). 

Day 0 vs. Day 28: In SED horses, plasma iohexol was significantly higher on Day 0 than on Day 28 at P3 and P5 (*p* = 0.04 and 0.05, respectively). There were no significant differences between Day 0 and Day 28 in EX horses (*p* = 0.23) (Figure 1).

#### 3.1.2. Supplemented Diet (Figure 2)

Day 0: In SED horses receiving the SUPP diet, there was a significant increase in plasma iohexol between P1 (0.51 ± 0.03 ug/mL) and P2 (0.87 ± 0.04 ug/mL) (*p* = 0.005), P3 (0.82 ± 0.06 ug/mL) (*p* = 0.02) and P4 (0.97 ± 0.09 ug/mL) (*p* < 0.001). EX horses demonstrated a significant increase in plasma iohexol between P1 (0.70 ± 0.15 ug/mL) and P3 (1.75 ± 0.19 ug/mL) (*p* = 0.01). Plasma iohexol was significantly higher in EX horses than in SED horses at P3 (SED: 0.82 ± 0.06 ug/mL; EX: 1.75 ± 0.19 ug/mL) (*p* < 0.001) (Figure 2). 

Day 28: In SED horses receiving the SUPP diet, there was no significant change in plasma iohexol at any time between P1 (0.49 ± 0.05 ug/mL) and P7 (0.70 ± 0.05 ug/mL) (*p* = 0.43). There was also no significant increase in plasma iohexol in EX horses at any time between P1 (0.87 ± 0.23 ug/mL) and P7 (0.56 ± 0.12 ug/mL) (*p* = 0.36)(Figure 2). 

#### 3.1.3. Day 0 and Day 28 in Supplemented and Control Diets

On Day 0, iohexol tended to be higher in SUPP than CO horses (*p* = 0.053). Overall iohexol was significantly elevated in EX horses at P2, P3, (*p* < 0.001) and P4 (*p* = 0.02) compared with P1, but there were no differences between treatment groups (Figure 2) 

On Day 28, iohexol was significantly higher overall in CO horses compared with SUPP horses (*p* = 0.008). Overall, iohexol was significantly higher at P3 than P1, but there were no significant differences between treatment groups (Figure 2). 

### 3.2. Systemic Inflammation

#### 3.2.1. Serum Amyloid A (SAA; Table 2)

##### Control Diet

Day 0: In SED horses receiving the CO diet, there was no significant change in SAA at any time between P1 (0.10 ± 0.1 μg/mL) and P7 (0.10 ± 0.1 μg/mL) (*p* = 0.78). There was also no significant change in EX horses in SAA between P1 (0.22 ± 0.16 μg/mL) and P7 (0.86 ± 0.56 μg/mL) (*p* = 0.70). Overall, SAA was significantly higher in EX than in SED horses (*p* = 0.01), but there were no significant differences between groups at any specific time point (Table 2). 

Day 28: In SED horses receiving the CO diet, there was no significant change in SAA at any time between P1 (0.0 ± 0.0 μg/mL) and P7 (0.10 ± 0.10 μg/mL) (*p* = 0.92). There was also no significant change in EX horses in SAA between P1 (0.15 ± 0.15 ug/mL) and P7 (0.20 ± 0.20 μg/mL) (*p* = 0.96). In horses receiving the CO diet, SED horses had significantly lower SAA than EX horses overall (*p* = 0.04), but there were no significant differences at individual time points (Table 2). 

##### Supplemented Diet

Day 0: In SED horses receiving the SUPP diet, there was no significant change in SAA at any time between P1 (0.33 ± 0.33 μg/mL) and P7 (0.15 ± 0.15 μg/mL) (*p* = 0.71). There was also no significant change in EX horses SAA between P1 (0.08 ± 0.08 μg/mL) and P7 (0.30 ± 0.30 μg/mL) (*p* = 0.70). There were no significant differences between SED and EX at any specific time point on Day 0 (Table 2). 

Day 28: In SED horses receiving the SUPP diet, there was no significant change in SAA at any time between P1 (0.17 ± 0.17 μg/mL) and P7 (0.35 ± 0.15 μg/mL) (*p* = 0.59). There was also no significant change in EX horses in SAA between P1 (0.35 ± 0.25 μg/mL) and P7 (1.00 ± 0.53 μg/mL) (*p* = 0.96). Overall, SAA was significantly higher in EX than in SED horses (*p* = 0.02), but there were no significant differences between groups at specific time points (Table 2).

##### Day 0 and Day 28 in Supplemented and Control Diets

On Day 0, there were no differences in SAA between SUPP and CO horses (*p* = 0.257). Overall, SAA was significantly higher in EX than SED horses (*p* = 0.015), primarily owing to significantly higher SAA in EX than SED horses in CO horses (*p* = 0.002) that was not observed in SUPP horses (*p* = 0.826) (Table 2). 

On Day 28, SAA was significantly higher overall in SUPP horses compared with CO horses (*p* = 0.01). There was no significant difference in SAA between EX and SED horses overall, but SAA was significantly higher in SED horses than EX horses in horses receiving the supplemented diet (*p* = 0.05) (Table 2).

#### 3.2.2. Lipopolysaccharide (LPS; Table 2)

##### Control Diet 

Day 0: In SED horses receiving the CO diet, there was no significant change in LPS at any time between P1 (2.10 ± 0.09 pg/mL) and P7 (2.13 ± 0.12 pg/mL) (*p* = 0.71). There was also no significant change in EX horses in LPS between P1 (2.18 ± 0.06 pg/mL) and P7 (2.21 ± 0.10 pg/mL) (*p* = 0.99). Overall, LPS was significantly higher in EX than in SED horses (*p* = 0.02), but there were no significant differences between SED and EX at any specific time point (Table 2). 

Day 28: In SED horses receiving the CO diet, there was no significant change in LPS at any time between P1 (2.1 ± 0.09 pg/mL) and P7 (2.1 ± 0.05 pg/mL) (*p* = 0.94). There was also no significant change in EX horses in LPS between P1 (2.14 ± 0.03 pg/mL) and P7 (2.10 ± 0.08 pg/mL) (*p* = 0.94). Overall, LPS was significantly higher in EX than in SED horses (*p* = 0.004), but there were no significant differences between groups at specific time points (Table 2). 

##### Supplemented Diet

Day 0: In SED horses receiving the SUPP diet, there was no significant change in LPS at any time between P1 (2.15 ± 0.04 pg/mL) and P7 (2.17 ± 0.04 pg/mL) (*p* = 0.91). There was also no significant change in EX horses LPS between P1 (2.06 ± 0.04 pg/mL) and P7 (2.13 ± 0.01 pg/mL) (*p* = 0.98). LPS was significantly higher in SED than EX horses (*p* = 0.03), but there were no significant differences between groups at specific time points (Table 2).

Day 28: In SED horses receiving the SUPP diet, there was no significant change in LPS at any time between P1 (2.20 ± 0.08 pg/mL) and P7 (2.18 ± 0.07 pg/mL) (*p* = 0.90). There was also no significant change in EX horses in LPS between P1 (2.06 ± 0.04 pg/mL) and P7 (2.06 ± 0.05 pg/mL) (*p* = 0.97). LPS was significantly higher in SED than EX horses overall (*p* < 0.001), as well as at P5 (*p* = 0.01) and P6 (*p* = 0.05) (Table 2). 

##### Day 0 and Day 28 in Supplemented and Control Diets

On Day 0, there were no differences in LPS between SUPP and CO horses (*p* = 0.346). There was also no significant difference between EX and SED horses overall (*p* = 0.268). LPS was significantly higher in EX than SED horses in the CO group (*p* = 0.003), but there were no significant differences in LPS between EX and SED horses in the SUPP group (*p* = 0.068) (Table 2).

On Day 28, there were no differences in LPS between SUPP and CO horses (*p* = 0.674). There was also no significant difference between EX and SED horses overall (*p* = 0.392). LPS was significantly higher in EX than SED horses in the CO group (*p* = 0.004) and significantly lower in EX than SED in the SUPP group (*p* < 0.001) (Table 2).

## 4. Discussion

The purpose of the current study was to quantify the effect of a dietary *A. oryzae* prebiotic on gastrointestinal permeability in horses challenged with combined transport and exercise stress. The main finding was that 28 days of supplementation with the *A. oryzae* prebiotic completely eradicated stress-induced gastrointestinal permeability in this group of horses.

We have previously demonstrated that the combination of transport and exercise stress model utilized in the current study produces gastrointestinal hyperpermeability and an increase in blood biomarkers that evidence transient, low-grade systemic inflammation [7]. Like our previous study, we report herein that 60 min of trailer transport immediately preceding half an hour of moderate-intensity exercise is a clear, reproducible model of gastrointestinal hyperpermeability. On Day 0 for both feeding groups, the stress model resulted in a significant uptick in the systemic appearance of orally administered iohexol that was not seen in unstressed controls. That this spike in the systemic appearance of iohexol was absent in stressed horses in the SUPP feeding group on Day 28 provides strong evidence for the role of *A. oryzae* prebiotic in protecting gastrointestinal barrier function in horses during stress. The mechanism for this blockade is not known but may be associated with an effect of *A. oryzae* prebiotic on the enteric microbiome. *A. oryzae* strongly increases the relative abundance of anti-inflammatory bacterial strains such as Bifidobacterium [18,24] and important fiber-degrading bacteria such as Ruminococcaceae [19]. Dietary provision of Bifidobacterium-based probiotics to obese humans results in a marked decrease in gastrointestinal hyperpermeability [25], which provides support for the hypothesis that *A. oryzae* prebiotic protects the enteric barrier from stress-induced hyperpermeability via its modulation of the gastrointestinal microbiome. This hypothesis should be tested in future studies.

When dietary groups were combined, there was an overall increase in SAA in response to our stress challenge, consistent with our previous study [7], but this effect was not observed when analyzing dietary groups individually. SAA is the major acute phase protein in the horse. While it is a highly sensitive indicator of an inflammatory event, it is not specific, and its production can be markedly increased in the presence of almost any inflammatory challenge [26]. The vast majority of SAA is produced by hepatocytes, but small amounts may also be produced by enterocytes [27]. Our small sample size, together with SAA fluctuations in both EX and SED groups that were unrelated to our stress challenge, likely contributed to the lack of statistical increase in SAA within groups. Consequently, the effect of *A. oryzae* prebiotic on this biomarker remains unknown. Owing to the highly plastic nature of SAA in vivo, future studies to evaluate the effects of the *A. oryzae* prebiotic on this outcome measure may benefit from controlled in vitro assessment of enterocyte-specific production of SAA [27].

The marked gastrointestinal hyperpermeability that was observed in the current study in EX horses in the control feeding group on Days 0 and 28 was not associated with a significant time-dependent increase in circulating LPS, and like SAA, this may have been due, at least in part, to our small sample size. But the overall serum LPS concentration of EX horses was significantly higher than SED horses. Surprisingly, however, serum LPS was significantly lower in EX than in SED horses for the *A. oryzae* feeding group. This result is probably not associated with the supplement because it was observed both on Day 0 (prior to beginning supplementation) and on Day 28, so instead is more likely an artifact of randomizing a small number of animals to the feeding groups. Furthermore, our maximum LPS concentration of 2.24 pg/mL in either feeding group is well within the reference interval for the normal flux of systemic LPS in healthy horses [26]. Future studies designed to detect the effect of the dietary *A. oryzae* prebiotic on the translocation of enteric LPS at levels expected to be associated with disease will require a stronger stress challenge such as non-steroidal anti-inflammatory drugs [2,27].

The current study had fewer animals in each treatment group than our previous study, which may have resulted in the current study being underpowered to detect the effects of stress and/or diet on SAA and LPS.

## 5. Conclusions

In conclusion, the data presented herein provide compelling evidence for a protective effect of *A. oryzae* prebiotic on stress-induced gastrointestinal hyperpermeability. This supplement may be a useful dietary ingredient for horses undergoing combined transport and exercise stress as a prevention for gastrointestinal hyperpermeability. Future studies should explore the effects of *A. oryzae* prebiotic on the equine gastrointestinal microbiome as a potential mode of action. 

## Figures and Tables

**Figure 1 animals-13-00951-f001:**
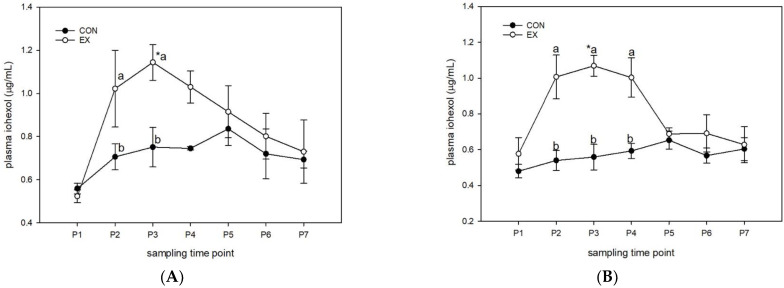
Plasma iohexol (µg/mL). Horses (n = 8) on the CO diet were administered an oral dose of iohexol (5.6% solution; 1 mL/kg BW) immediately prior to one hour of trailer transport followed by 30 min of moderate-intensity exercise (EX) on Day 0 (Panel (**A**)) and Day 28 (Panel (**B)**). Unchallenged horses (SED; n = 8) were maintained in stalls as controls and sampled at the same time intervals. Blood was sampled prior to iohexol administration (P1), immediately after trailering (P2), immediately after exercise (P3), and 1 (P4), 2 (P5), 4 (P6), and 8 h post-exercise (P7). * denotes significant change from baseline within a group; different letters denote significant differences between groups at a given time point.

**Figure 2 animals-13-00951-f002:**
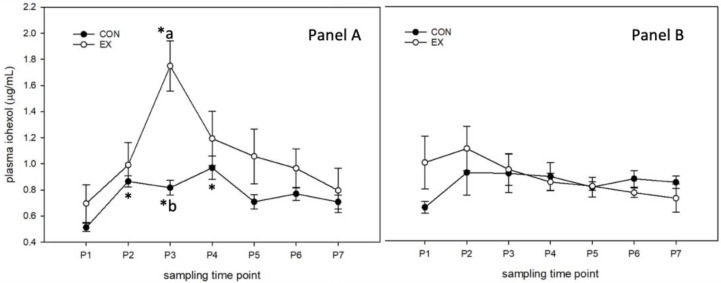
Plasma iohexol (µg/mL). Horses (n = 8) on the SUPP diet were administered an oral dose of iohexol (5.6% solution; 1 mL/kg BW) immediately prior to one hour of trailer transport followed by 30 min of moderate-intensity exercise (EX) prior to supplementation (Panel **A**) and after 28 days of supplementation (Panel **B**). Unstressed horses (SED; n = 8) were maintained in stalls as controls and sampled at the same time intervals. Blood was sampled prior to iohexol administration (P1), immediately after trailering (P2), immediately after exercise (P3), and 1 (P4), 2 (P5), 4 (P6), and 8 h post-exercise (P7). * denotes significant change from baseline within a group; different letters denote significant differences between groups at a given time point.

**Table 1 animals-13-00951-t001:** Horse ration pellets (12%).

Nutritional Analysis (as Fed)	
Crude Protein	11.99%
Lysine	0.49%
Crude Fat	3.60%
Crude Fiber	9.34%
Dry Matter	89.09%
Calcium (total)	0.74%
Phosphorus (total)	0.52%
Sodium	0.35%
Chloride	0.53%
Potassium	0.72%
Magnesium	0.29%
Sulfur	0.15%
Iron	110.71 mg/kg
Manganese	150.26 mg/kg
Zinc	182.22 mg/kg
Copper	36.46 mg/kg
Iodine	0.98 mg/kg
Selenium	0.40 mg/kg
Cobalt	3.00 mg/kg
Vitamin A	12.53 KIU/kg
Vitamin D3	2.51 KIU/kg
Vitamin E	200.00 KIU/kg
Biotin	1640.00 mcg/kg
D.E. Horse	3.06 Mcal/kg
TDN Horse	72.31%
Starch	23.21%

**Table 2 animals-13-00951-t002:** Serum amyloid A (SAA; ug/mL ± SEM) and Lipopolysaccharide (LPS; pg/mL ± SEM) in horses challenged with combined transport and exercise (EX) or maintained as unchallenged controls (SED), before (Day 0) or 28 days after (Day 28) receiving a diet containing an *Aspergillus oryzae* prebiotic (SUPP) or a control diet (CO).

	SAA						LPS					
	Day 0											
	CO			SUPP			CO			SUPP		
	SED ^a^	EX ^b^	*p ***	SED	EX	*p ***	SED ^a^	EX ^b^	*p ***	SED ^b^	EX ^a^	*p ***
P1	0.10 ± 0.10	0.22 ± 0.16	0.9	0.33 ± 0.33	0.08 ± 0.08	0.8	2.10 ± 0.09	2.18 ± 0.06	1.0	2.15 ± 0.04	2.06 ± 0.04	1.0
P2	0.03 ± 0.03	0.12 ± 0.10	0.9	0.03 ± 0.03	0.00 ± 0.00	0.8	2.09 ± 0.06	2.23 ± 0.06	1.0	2.14 ± 0.02	2.09 ± 0.02	1.0
P3	0.00 ± 0.00	0.52 ± 0.24	0.9	0.10 ± 0.10	0.23 ± 0.17	0.8	2.14 ± 0.06	2.22 ± 0.10	1.0	2.16 ± 0.05	2.13 ± 0.04	1.0
P4	0.00 ± 0.00	0.46 ± 0.30	0.9	0.20 ± 0.20	0.03 ± 0.03	0.8	2.10 ± 0.06	2.24 ± 0.13	1.0	2.19 ± 0.03	2.11 ± 0.05	1.0
P5	0.03 ± 0.03	0.74 ± 0.43	0.9	0.18 ± 0.18	0.10 ± 0.10	0.8	2.08 ± 0.04	2.19 ± 0.05	1.0	2.15 ± 0.04	2.10 ± 0.05	1.0
P6	0.03 ± 0.03	0.78 ± 0.45	0.9	0.08 ± 0.08	0.00 ± 0.00	0.8	2.09 ± 0.04	2.24 ± 0.08	1.0	2.18 ± 0.01	2.14 ± 0.05	1.0
P7	0.10 ± 0.10	0.86 ± 0.56	0.9	0.15 ± 0.15	0.30 ± 0.30	0.8	2.13 ± 0.12	2.21 ± 0.10	1.0	2.17 ± 0.04	2.13 ± 0.01	1.0
*p **	0.78	0.70		0.92	0.96		0.71	0.99		0.91	0.98	
	Day 28											
	CO			SUPP			CO			SUPP		
	SED ^a^	EX ^b^	*p ***	SED	EX	*p ***	SED ^a^	EX ^b^	*p* **	SED ^b^	EX ^a^	*p ***
P1	0.00 ± 0.00	0.15 ± 0.15	1.0	0.17 ± 0.17	0.35 ± 0.25	0.8	2.10 ± 0.09	2.14 ± 0.03	0.68	2.20 ± 0.08	2.06 ± 0.04	0.11
P2	0.00 ± 0.00	0.15 ± 0.12	1.0	0.07 ± 0.07	0.50 ± 0.50	0.8	2.04 ± 0.04	2.15 ± 0.05	0.20	2.15 ± 0.08	2.04 ± 0.04	0.22
P3	0.00 ± 0.00	0.23 ± 0.17	1.0	0.23 ± 0.15	0.47 ± 0.37	0.8	2.07 ± 0.07	2.17 ± 0.05	0.26	2.13 ± 0.05	2.03 ± 0.03	0.26
P4	0.00 ± 0.00	0.08 ± 0.05	1.0	0.07 ± 0.07	1.33 ± 1.00	0.8	2.00 ± 0.05	2.17 ± 0.04	0.05	2.16 ± 0.07	2.05 ± 0.05	0.20
P5	0.03 ± 0.03	0.23 ± 0.23	1.0	0.03 ± 0.03	0.63 ± 0.41	0.8	2.04 ± 0.05	2.19 ± 0.07	0.10	2.24 ± 0.04	2.02 ± 0.03	0.01
P6	0.03 ± 0.03	0.35 ± 0.22	1.0	0.13 ± 0.09	0.45 ± 0.45	0.8	2.08 ± 0.08	2.18 ± 0.08	0.25	2.24 ± 0.10	2.06 ± 0.04	0.05
P7	0.10 ± 0.10	0.20 ± 0.20	1.0	0.35 ± 0.15	1.00 ± 0.53	0.8	2.07 ± 0.05	2.10 ± 0.08	0.68	2.18 ± 0.07	2.06 ± 0.05	0.17
*p **	0.92	0.96		0.71	0.70		0.94	0.94		0.90	0.97	

P1: before iohexol administration; P2: immediately after 1 h of trailer transport; P3 immediately after 30 min of moderate-intensity exercise; P4: after 1 h of exercise recovery; P5: after 2 h of exercise recovery; P6: after 4 h of exercise recovery; P7: after 8 h of e after 8 h of exercise recovery; different lower-case letters denote significant differences between SED and EX overall; *p* * = *p*-values for one-way ANOVA within treatment group; *p* ** *p*-values for 2-way ANOVA time × treatment group within diet.

## Data Availability

All processed data, final data, and figures will be available for sharing within the University of Guelph via the Agri-environmental research data repository. For access outside the University, users can contact the PI for special access to data.

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
