# Peer review of "Dietary Fermentation Product of Aspergillus Oryzae Prevents Increases in Gastrointestinal Permeability (‘Leaky Gut’) in Horses Undergoing Combined Transport and Exercise"

_animals, 2023, doi:10.3390/ani13050951_

Round 1
Reviewer 1 Report
Dear authors, thank you so much for your manuscript. I think that this topic is of particular interest and your work can add more comprehension on the mechanism related to the presence of leaky gut in horses. I think that your work is well-designed and written. However, it is needed to implement more of the materials and methods section before it is accepted for publication.
Check for spelling and space between words because there are several formatting errors.
I suggest underlining in Introduction that – as stated by Stewart (reference 1) – that one of the main causes of leaky gut in equine species is related to feeding horses an unbalanced diet characterised by high amounts of starch. Here is the link to the paper in which a research team evaluated the effects of a high starch diet on intestinal permeability (https://doi.org/10.1111/jpn.13643). Moreover, I suggest implementing the section on line 71. In fact, high starch diets are responsible for gastrointestinal dysbiosis also in equine species and this dysbiosis was found to negatively impact on horse gut environment (e.g. VFAs, https://doi.org/10.1186/s12917-022-03289-2) and on histological and morphometrical parameters of the intestinal wall (https://doi.org/10.1186/s12917-022-03433-y)
Line 53 – maybe it should be added in 18 Arabian horse…
Lin 68 – LGS
Line 92 – correlate
Line 105 – Please, provide the grams of pelleted feed supplied… “one cup” is not adequate
Table 1. I’m sorry but I don’t understand what you mean by “Guaranteed analysis for 12% Horse Ration Pellets”. Moreover, what does it mean “min” and “actual”? The data you have reported in the Table are on the dry matter or as-fed? Could you provide the complete proximate analysis of the pelleted feed and also the list of ingredients?
More information on the management of horses involved in the study must be added:
- line 104: which kind of hay? Meadow?
- when and how the pelleted feed was supplied? Once a day? Which meal? Singularly?
- When and how the SUPP was supplied? Once a day? Which meal? Singularly? It was a powder, a pellet or what? Did you mix with the pelleted feed?
- There were feeding wastes? Did you monitor them?
Line 148 – I would describe sample analyses 2.4-2.5-2.6-2.7 in only one paragraph.
Figure 1 – add Panel A and Panel B to the figure (as you did for Figure 2).
Table 2 – p values should be added to the Table.
Since exercising horses are often fed high amounts of starch in their diet, I suggest discussing that the diet you provided to your horses did not overcome the recommended safe level of starch of <2g/BW horse/meal – for this reason, more information about the diet should be provided in the m&m section.
Author Response
I suggest underlining in Introduction that – as stated by Stewart (reference 1) – that one of the main causes of leaky gut in equine species is related to feeding horses an unbalanced diet characterised by high amounts of starch. Here is the link to the paper in which a research team evaluated the effects of a high starch diet on intestinal permeability (https://doi.org/10.1111/jpn.13643).
- Thank you for this suggestion. We have added this to the introduction. Please see Line 62.
Moreover, I suggest implementing the section on line 71. In fact, high starch diets are responsible for gastrointestinal dysbiosis also in equine species and this dysbiosis was found to negatively impact on horse gut environment (e.g. VFAs, https://doi.org/10.1186/s12917-022-03289-2) and on histological and morphometrical parameters of the intestinal wall (https://doi.org/10.1186/s12917-022-03433-y)
- We have added this point into the introduction. Please see lines 81-82.
Line 53 – maybe it should be added in 18 Arabian horse…
- I’m not sure to what the reviewer is referring. 18 Arabian horses is already stated here..? Please clarify, and we will be happy to address the comment.
Lin 68 – LGS
- We have made this correction
Line 92 – correlate
- Thank you. We have made this correction.
Line 105 – Please, provide the grams of pelleted feed supplied… “one cup” is not adequate
Table 1. I’m sorry but I don’t understand what you mean by “Guaranteed analysis for 12% Horse Ration Pellets”. Moreover, what does it mean “min” and “actual”?
- A ‘Guaranteed Analysis’ is a tag that is placed onto a feed bag for the purpose of regulatory compliance. The GA provides information pertaining to nutrient composition as well as any voluntary label claims. As a minimum the GA must include information pertaining to protein, fat, fibre and moisture, but most feed manufacturers also include additional information pertaining to selected micronutrients and energy.
- Min, Actual: This is a very common convention on feed tags and complies with laws governing feed manufacturing. ‘min’ refers to the minimum amount of a nutrient that is present in the feed. The actual amount may in fact be higher but must not be lower. ‘actual’ is the actual amount of a nutrient, which does not vary from batch to batch.
The data you have reported in the Table are on the dry matter or as-fed?
- These data are as fed. I have added this information to the Table legend.
Could you provide the complete proximate analysis of the pelleted feed and also the list of ingredients?
- In Table 1, the Guaranteed Analysis feed tag information has been replaced with the complete nutritional analysis. The feed ingredients list is not provided by the manufacturer so has not been included here.
More information on the management of horses involved in the study must be added:
- line 104: which kind of hay? Meadow?
- Timothy hay. This information has been added to line 130.
- when and how the pelleted feed was supplied? Once a day? Which meal? Singularly?
- This information has been added to lines 130-131
- When and how the SUPP was supplied? Once a day? Which meal? Singularly? It was a powder, a pellet or what? Did you mix with the pelleted feed? There were feeding wastes? Did you monitor them?
- This information has been added to lines 137-141
Line 148 – I would describe sample analyses 2.4-2.5-2.6-2.7 in only one paragraph.
- This change has been made.
Figure 1 – add Panel A and Panel B to the figure (as you did for Figure 2).
- This correction has been made
Table 2 – p values should be added to the Table.
- P-values have been added
Since exercising horses are often fed high amounts of starch in their diet, I suggest discussing that the diet you provided to your horses did not overcome the recommended safe level of starch of <2g/BW horse/meal – for this reason, more information about the diet should be provided in the m&m section.
- The horses used in the study were sedentary (please see lines 131-132) and only exercised sporadically as part of this study. They were consuming a diet comprised mainly of hay, with just a small amount (250g) of a concentrate pellet provided as a vehicle for the test supplement. The additional information on the diet has been added to the methods section as suggested.
Reviewer 2 Report
Overall Comments:
The research objective was to evaluate the effect of a prebiotic Aspergillus oryzae product (SUPP) on stress-induced gastrointestinal permeability. There was initial enthusiasm for the presented work; however, that was diminished by the statistical analysis and presentation of results which led the authors to inappropriately derive conclusions comparing dietary treatment and sampling timepoints.
Major Concerns:
- In the simple summary, it is unclear when exercise and transport occurred relative to the supplementation period. Did it occur on days 0 and 28 like iohexol administration?
- A sample size of 2 is drastically small. The crossover design improves this slightly.
- The abstract is clearer as to when the iohexol and exercise/trailering occurred in the experimental timeline. This needs to be reflected in the simple summary.
- The use of CO and CON makes it confusing to differentiate between the dietary control and the challenge control groups.
- Were horses that exercised in the morning in period one exercised at the same time in period 2?
- What were the inter- and intra-assay variations for the assays used?
- Details on sample analysis are lacking.
- Period is not included in the statistical model which is a flaw of the data analysis and is likely because the degrees of freedom are too low with a sample size of 2.
- The only statistical results presented are between control and exercise challenged groups within a dietary treatment and within a sampling timepoints. There are no statistical comparisons between day 0 and day 28 or between dietary treatments. Therefore, it is inappropriate to compare the two sampling timepoints or draw conclusions about dietary treatments without the statistical data to support the comparisons.
- Table 2: P-value are excluded from the table making it impossible for the reader to develop their own interpretation of the results. Further, the setup of the table is not well developed.
- Discussion – the lack of significance in SAA and LPS is likely due to the small sample size.
Minor Concerns:
Lines 38-39: There is redundancy in “on day 0” repeated in the sentence.
Line 174: include a space before SD
Author Response
Major Concerns:
In the simple summary, it is unclear when exercise and transport occurred relative to the supplementation period. Did it occur on days 0 and 28 like iohexol administration?
- this has been clarified in the simple summary. Please see line 15.
A sample size of 2 is drastically small. The crossover design improves this slightly.
The abstract is clearer as to when the iohexol and exercise/trailering occurred in the experimental timeline. This needs to be reflected in the simple summary.
- thank you. This change has been made.
The use of CO and CON makes it confusing to differentiate between the dietary control and the challenge control groups.
- we have changed ‘CON’ to ‘SED’ throughout the manuscript
- Were horses that exercised in the morning in period one exercised at the same time in period 2?
- This information has been added to the methods, line 158-159
- What were the inter- and intra-assay variations for the assays used?
- CVs have been added to the methods, lines 200, 216-217.
- Details on sample analysis are lacking.
- The methods for sample analysis were identical to those reported in our earlier publication (McGilloway et al. 2022), which is why we provided only brief description here and provided the reference for further details.
- Period is not included in the statistical model which is a flaw of the data analysis and is likely because the degrees of freedom are too low with a sample size of 2. The only statistical results presented are between control and exercise challenged groups within a dietary treatment and within a sampling timepoints. There are no statistical comparisons between day 0 and day 28 or between dietary treatments. Therefore, it is inappropriate to compare the two sampling timepoints or draw conclusions about dietary treatments without the statistical data to support the comparisons.
- Day 0 vs 28 comparisons according to diet for all outcomes have added (please see lines 321-327, 410-418, and 480-488). We have also made adjustments to the discussion which arose from these new analyses (lines 606-608,612-615, 621)
- Table 2: P-value are excluded from the table making it impossible for the reader to develop their own interpretation of the results. Further, the setup of the table is not well developed.
- p-values have been added to Table 2.
- Discussion – the lack of significance in SAA and LPS is likely due to the small sample size.
- We have added a comment to address this in line 612,
Minor Concerns:
Lines 38-39: There is redundancy in “on day 0” repeated in the sentence.
- Thank you for noticing this – we have corrected this error
Line 174: include a space before SD
- Thank you. Done.
Round 2
Reviewer 1 Report
Thank you for having answered to my previous comments. I just have another consideration.. please, check table 2 since it seems that p** are lacking for Day 28
Author Response
Thank you for noticing that oversight. It has now been corrected in the revised version.
Reviewer 2 Report
Authors have greatly improved the presentation of methodology, results, and conclusions.
Author Response
Thank you for the time you have taken to provide feedback on our manuscript.